# Formulating AutoML as a Variable-Length Optimization Problem: A Tree of Thought Approach with LLM-Driven Code Generation

## Abstract

Recent advancements in machine learning have created a demand for automated systems that enable efficient development and deployment of machine learning applications. Traditional Automated Machine Learning (AutoML) approaches often rely on fixed pipeline structures, which limit adaptability to diverse task complexities. In this paper, we introduce a novel formulation of AutoML as a variable-length optimization problem, allowing for dynamic adjustment of model architectures based on task requirements. To effectively navigate the expanded search space of variable-length models, we employ the Tree of Thoughts (ToT) method combined with Large Language Models (LLMs). This framework utilizes a sequential decision-making process, allowing models to be incrementally constructed by evaluating prior outcomes. Additionally, LLMs automatically generate the code corresponding to each decision, transforming model configurations into executable pipelines and reducing manual intervention. Our approach enhances efficiency by focusing on promising pathways and improves transparency by explicitly showcasing how each decision contributes to the overall optimization. Experiments conducted on diverse datasets, including OpenML and clinical tasks, demonstrate that our method outperforms traditional AutoML systems, delivering superior model performance and better adaptability across different task complexities.

## 1. Introduction

The recent substantial progress in machine learning (ML) and deep learning has created a significant demand for hands-free systems that enable both developers and ML novices to efficiently build and deploy machine learning applications (Baratchi et al., 2024). Since different datasets often require unique ML pipelines, this demand has driven the development of automated machine learning (AutoML) (Santu et al., 2022). AutoML aims to streamline and automate the process of designing, training, and optimizing machine learning models, effectively reducing the need for deep domain expertise and manual tuning. Its applications span various domains, from healthcare (Waring et al., 2020), biology (Valeri et al., 2023; Yu et al., 2024), and drug discovery (Turon et al., 2023), underscoring its critical role in unlocking the full potential of machine learning.

Current mainstream AutoML systems, like Auto-WEKA (Thornton et al., 2013), H2O (LeDell & Poirier, 2020), and Auto-sklearn (Feurer et al., 2019), typically rely on fixed pipeline structures, as shown in Figure 1(a). These pipelines follow a linear sequence of steps—data cleaning, feature transformation, feature selection, and modeling—where each step selects an operation from a predefined candidate pool. The search space is static, with model configurations limited by fixed parameters like depth, number of nodes, and layers. This approach assumes that tasks share similar complexity levels, allowing uniform models to be applied across different tasks. However, this fixed-structure paradigm may lead to inferior pipeline performances for complex datasets—characterized by varying data distributions, objectives, and scales, for example, multiple preprocessing steps or feature trnasformation (Hollmann et al., 2023; Mumuni & Mumuni, 2024). Optimal configurations often need structural flexibility and variable complexity to meet specific task demands. Fixed-length models, by assuming uniform task complexity, may not fully account for the varying needs of both simple and complex tasks. Efforts to improve flexibility, such as Olson & Moore's introduction of a dataset duplicator, allow for parallel processing paths within pipelines. These paths can be merged using the feature union operator, which combines multiple datasets or processing paths. While this enhances flexibility, the overall pipeline depth and complexity are still constrained by predefined limits, restricting the adaptability needed for more diverse or demanding tasks.

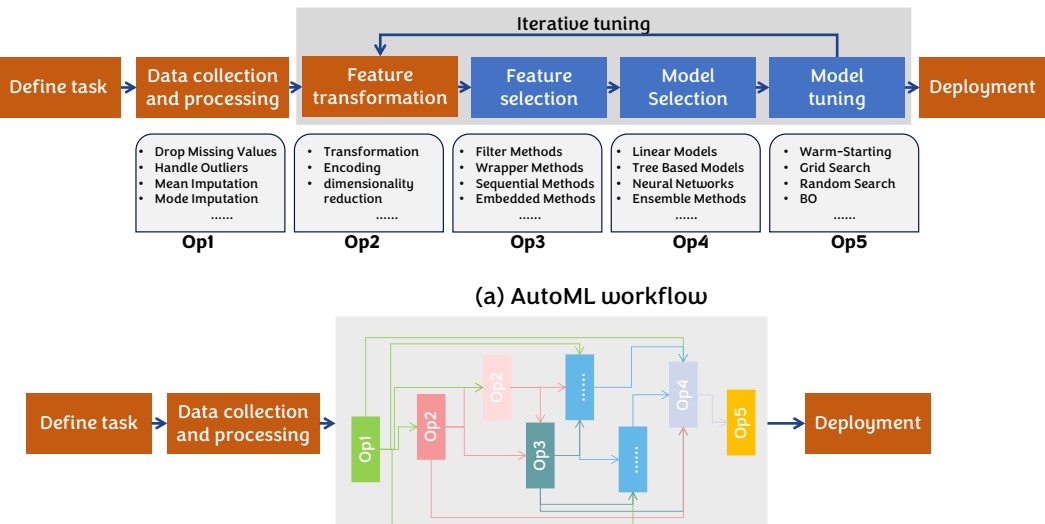

(a) AutoML workflow

(b) AutoML workflow with variable-length

Figure 1: Comparison between the original AutoML workflow and the enhanced variable-length AutoML workflow. In (b), different colors represent different operations, and the ellipses indicate that intermediate connections may include other operations.

To better adapt the model to a variety of tasks, we propose formulating the AutoML problem as a *variable-length optimization problem*, as shown in Figure 1 (b). Unlike fixed structures, this approach allows models to dynamically adjust to the specific demands of each task, whether simple or complex. Flexibility is critical in optimization since overly complex models can lead to overfitting in simple tasks, while insufficiently complex models may underfit more complex tasks (Bejani & Ghatee, 2021; Bian & Priyadarshi, 2024). Thus, the model's structure should be adaptable, allowing simpler tasks to be handled with shallower models, and complex tasks with deeper, more sophisticated networks.

Additionally, traditional methods that explore the entire framework and hyperparameter space simultaneously are inefficient due to the sheer size of the search space, which makes exhaustive exploration computationally prohibitive. A sequential decision-making approach (Barto et al., 1989) is better suited to this problem, as it allows models to be constructed step by step, dynamically adjusting complexity in response to task demands (Jaâfra et al., 2019). Furthermore, large language models (LLMs), equipped with vast knowledge across various domains (Brown et al., 2020), possess strong reasoning (Plaat et al., 2024), decision-making (Harte et al., 2023), and coding capabilities (Wang et al., 2024) . This allows them to understand task requirements and tailor model structures accordingly, making them ideal for navigating the expanded search space of variable-length models.

To efficiently solve this variable-length optimization problem, we employ the *Tree of Thoughts (ToT)* method using LLMs (Yao et al., 2023). The ToT method handles model structure selection through a *flexible sequential decision-making process*, where the model is incrementally constructed by evaluating the outcomes of each decision. This process enhances efficiency by focusing on promising pathways and pruning less viable ones early in the search space, reducing unnecessary computation. Moreover, this method guarantees *transparency* by explicitly demonstrating how each decision contributes to the model's overall optimization, as illustrated in Figure 2(a). Starting from basic transformations like Mean Imputation and Log Transformation, the ToT method refines the model step by step, selecting the most impactful preprocessing steps and tuning key model parameters. Furthermore, LLMs automatically generate code for each decision, transforming model configurations into executable pipelines, reducing manual intervention and streamlining the process.

We conducted experiments on diverse datasets, including OpenML and clinical tasks, to evaluate the efficacy of our approach. The results demonstrate that our method achieves superior model performance, outperforming traditional AutoML methods in terms of accuracy and adaptability to different task complexities, proving the flexibility, transparency, and effectiveness of the ToT method.

**Contributions.** We summarize the contributions of our paper as follows:

- We introduce a flexible AutoML framework by reformulating it as a variable-length optimization problem, enabling dynamic adjustments to model structure based on task complexity.
- We propose the Tree of Thoughts (ToT) method with Large Language Models (LLMs) to efficiently navigate the expanded search space of variable-length models, enhancing both search efficiency and transparency in the decision-making process.
- Our experiments demonstrate that the combination of variable-length optimization and the ToT method outperforms traditional AutoML methods in terms of model performance and computational efficiency across diverse tasks.

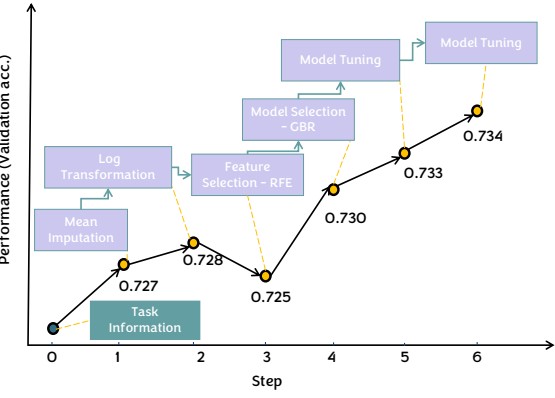

(a) The step-wise evolutionary path of using our method to solve the CMC task.

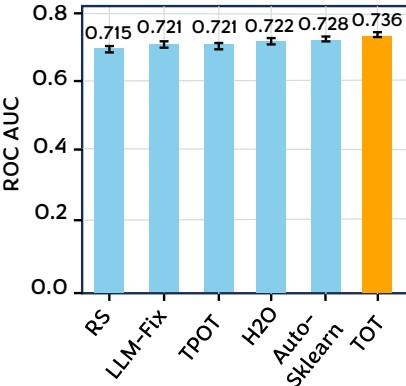

(b) Classification results for the CMC task.

Figure 2: Comparison of the evolutionary path and classification results for the CMC task in OpenML using various AutoML methods. The results in (b) highlight the superior performance of our method compared to state-of-the-art AutoML approaches.

## 2. PROBLEM DEFINITION

In this work, we formulate the AutoML process as a variable-length optimization problem. The objective is to identify the optimal sequence of operations that maximizes the model's performance. Let $n$ denote the maximum number of operations available in a given AutoML pipeline. The optimization problem can be defined as minimizing an objective function $f(x_1, x_2, \ldots, x_{n-1}, x_n)$, where each $x_i$ represents a specific operation selected at stage $i$ of the pipeline.

$$
\begin{aligned}
\text{Minimize} \quad & f(x_1, x_2, \ldots, x_{n-1}, x_n) \\
\text{subject to} \quad &
\begin{cases}
x_1 \in \text{Op1}, \\
x_i \in \{\text{Op1}, \text{Op2}, \text{Op3}\}, \quad \text{for } i = 2, \ldots, n-2, \\
x_{n-1} \in \text{Op4}, \\
x_n \in \text{Op5}
\end{cases}
\end{aligned}
\tag{1}
$$

where Op1 to Op5 represent operation set for data processing, feature transformation, feature selection, model selection, and model tuning, respectively. The constraints ensure that specific operations are chosen at appropriate stages of the pipeline. For instance, $x_1 \in \text{Op1}$ corresponds to an operation in the data processing stage, while $x_{n-1} \in \text{Op4}$ pertains to the model selection stage, and so forth. This formulation allows for a comprehensive and flexible exploration of possible AutoML pipelines, ensuring that the selected sequence of operations is tailored to the specific characteristics of the data and task at hand.

## 3. OPTIMIZATION WITH TOT

In this section, we present our method for optimizing AutoML problem defined in Section 2 using a Tree-of-Thought (ToT) approach facilitated by LLMs. Our method, illustrated in Figure 3,

comprises several four stages: prompt preparation, pipeline generation, program implementation, and iterative refinement. The process begins with the preparation of a comprehensive prompt that includes all necessary information about the dataset and task. This prompt guides the LLM in recommending the next operation to be added to the AutoML pipeline. After each step, the LLM generates the corresponding Python code for the newly selected operation. The updated pipeline is then tested on the training data to evaluate its performance at each stage. Underperforming pipelines are filtered out early, enabling the optimization process to focus on more promising candidates. The remaining pipelines undergo iterative refinement using the ToT approach, where additional steps are incrementally added based on the current pipeline's performance, continuing until a predefined termination condition is met.

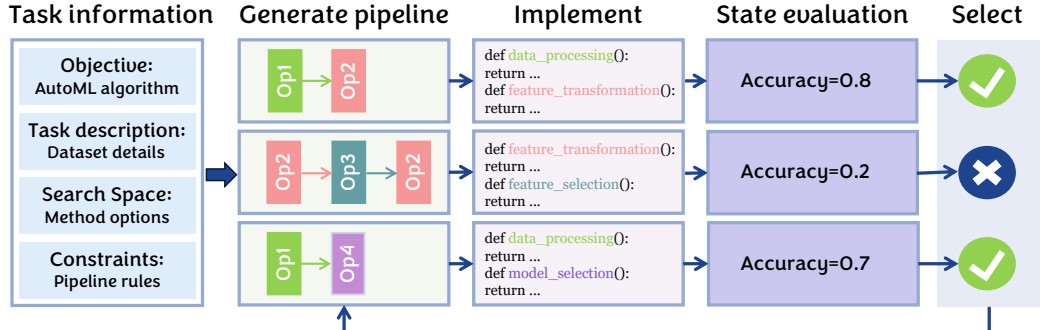

Figure 3: An overview of the our AutoML method. The process starts by preparing task-specific information, followed by: 1) Incrementally recommending the next operation in the pipeline using an LLM, 2) Translating each updated pipeline into executable Python code, 3) Evaluating the performance of each pipeline, and 4) Filtering the most promising candidates for further optimization.

## 3.1 GENERATING AUTOML PIPELINES

The first step in our method involves generating candidate AutoML pipelines incrementally, using a ToT approach with the help of LLMs. To guide the LLM effectively, we prepare a comprehensive prompt that includes the following components:

**Objective:** Assist in generating an AutoML pipeline for the dataset dataset. The objective is to recommend operations step-by-step for constructing an AutoML pipeline, based on the given search space, constraints, and requirements. These recommendations should cover data preprocessing, feature transformation, feature selection, model selection, and model tuning, while adhering to the provided constraints.

**Task Information:** Provide details on the training dataset, including the number of samples and features. Specify the feature types and indicate the percentage of missing values. Clarify whether the task is a classification or regression problem.

**Search Space:** Outline the available methods for each stage of the AutoML process, including options for preprocessing, transformation, selection, and model tuning.

**Constraints and Requirements:** (1) The first step in the pipeline must be a data processing method. (2) Feature transformation and feature selection must be completed before proceeding to model selection and model tuning. (3) Model selection and model tuning must occur before the final model training. (4) The total number of steps in the pipeline must not exceed max step.

To enhance the quality of the generated pipelines, we provide few-shot demonstrations—example tasks with corresponding pipelines. These examples help the LLM understand the expected format and components of effective pipelines.

The prompt also includes a **Step-by-Step Reasoning** section to encourage the LLM to perform a chain-of-thought analysis: (1) *Understanding the Dataset:* Analyze the dataset by examining its characteristics, including feature types and missing values. (2) *Identifying Challenges:* Determine the primary challenges based on the dataset's characteristics and the task type (classification or

regression). (3) *Applying Constraints:* Ensure that the operations adhere to the specified **constraints** and maintain the required sequence.

Finally, the LLM is asked to **Recommend the Next Step** for the current pipeline, ensuring that each new operation logically extends from the previous ones while aligning with the dataset's specific characteristics and needs.

## 3.2 IMPLEMENTING PYTHON PROGRAMS FROM PIPELINE

After generating the candidate pipelines, we proceed to implement them by prompting the LLM to generate corresponding Python code. Each pipeline is translated into a Python program capable of processing the dataset and producing performance metrics. We test each program on the training data to evaluate its performance. If a program fails to execute correctly or does not handle all training examples successfully, we engage in an iterative refinement process: (1) *Error Analysis:* Examine the execution results, including error messages and discrepancies between expected and actual outputs. (2) *Prompting for Revisions:* Prompt the LLM to revise the program based on the identified issues, providing specific feedback. (3) *Re-testing:* Test the revised program on the training data. This iterative process continues until the program executes successfully or a predefined number of iterations is reached. This approach leverages established techniques in code repair and debugging (Rahman et al., 2021). If, after several iterations, the program still does not perform satisfactorily, we manually coding the pipelines.

## 3.3 PERFORMANCE EVALUATION

To evaluate the effectiveness of each operation sequence in the pipelines—especially when specific model recommendations are absent—we employ a heuristic-based approach. This method ensures a robust assessment by introducing diversity and thorough exploration of potential models. The evaluation process involves the following steps: (1) *Model Selection:* Randomly select $n$ models from a predefined model library. This diversity allows us to cover a broad range of models that might be suitable for the dataset. (2) *Hyperparameter Tuning:* Each selected model undergoes hyperparameter tuning to identify the optimal configuration. Techniques such as grid search or random search are used to explore various parameter settings and enhance model performance. (3) *Performance Assessment:* Assess the performance of each model using relevant metrics—accuracy for classification tasks or mean squared error for regression tasks. (4) *Evaluation Score:* Select the highest performance score among the models, and use this score as the evaluation value for the current operation sequence. Mathematically, the evaluation function $f(\mathbf{x})$ for an operation sequence $\mathbf{x}$ is defined as:

$$f(\mathbf{x}) = \max_{i=1,2,\ldots,n} P(M_i(\mathbf{x})) \tag{2}$$

where $P(M_i(\mathbf{x}))$ denotes the performance of model $M_i$ after hyperparameter tuning.

## 3.4 REDUCING THE NUMBER OF CANDIDATE PIPELINES

After evaluating all candidate AutoML pipelines, we rank them by their performance and filter out the top-performing ones. This process effectively reduces the number of candidates, allowing us to focus on the most promising solutions, thereby improving optimization efficiency and the final model's performance.

## 4. EXPERIMENT SETUP

### 4.1 EVALUATED TASKS

We evaluate our approach on two distinct datasets: OpenML (Vanschoren et al., 2013; Hollmann et al., 2023) and clinical datasets (Arasteh et al., 2023; Wenzel et al., 2019). These datasets present a wide array of challenging tasks across various domains and feature counts, enabling a thorough assessment of our method's capabilities. Table 6 and Table 7 summarize the tasks and their corresponding datasets.

**OpenML Datasets:** We utilize a diverse selection of small datasets from OpenML, each featuring descriptive names and excluding those with numbered feature names (Hollmann et al., 2023). These datasets cover a broad spectrum of task types, including classification problems with feature counts ranging from 4 to 21 and sample sizes between 69 and 2,000. Each dataset includes a task description that provides context for our method.

**Clinical Datasets:** In addition to OpenML datasets, we assess our approach on clinical datasets that present unique challenges associated with medical data (Arasteh et al., 2023). These datasets, sourced from recent studies, encompass a variety of tasks, including diagnosing metastatic diseases and hereditary hearing loss. Each task is characterized by a wide range of feature counts (from 10 to 1,874) and distinct training and test set distributions, highlighting the complexity and specificity of real-world medical scenarios. Moreover, we include a task involving image classification for Parkinson's disease (Wenzel et al., 2019), which introduces an additional layer of complexity due to the nuanced visual patterns that must be discerned to accurately diagnose the condition.

### 4.2 EVALUATION PROTOCOL

For each dataset, we evaluate our method using 5 repetitions, each with a different random seed and train-test split to reduce the variance stemming from these splits (Hollmann et al., 2023). This approach ensures robustness by mitigating any bias introduced through specific data partitioning. In our method, the maximum step length is set to 6, and at each step, 3 solutions are generated and evaluated, with 3 solutions retained for the subsequent step. The GPT-4 model is used to assist in recommendation and coding throughout the optimization process. The stopping condition for the algorithm is consistent across methods, using a time limit of 1 hours per dataset to ensure fairness between methods, including peer AutoML frameworks.

## 5. EXPERIMENTAL RESULTS

In this section, we present a comprehensive evaluation of our proposed method, focusing on the comparison between variable-length and fixed-length optimization approaches, benchmarking against state-of-the-art AutoML frameworks, and conducting a detailed parameter sensitivity analysis.

### 5.1 HOW DOES VARIABLE-LENGTH OPTIMIZATION PERFORM COMPARED TO FIXED-LENGTH OPTIMIZATION

In this section, we investigate the performance differences between variable-length and fixed-length optimization approaches, aiming to ascertain if variable-length optimization offers improvements in efficiency compared to traditional fixed-length methods. We compare the performance of various optimization algorithms, including Random Search (RS), Bayesian Optimization (BO), LLM for fixed-length approaches, and our proposed `TOT` for variable-length. Each method was tested under both fixed (`RS-F`, `BO-F`, `LLM-F`) and variable (`RS-V`, `BO-V`, `TOT`) conditions. ROC AUC scores were utilized to measure the quality of solutions found across diverse classification tasks.

**Results on OpenML Datasets:** Table 1 shows the performance of each optimization method under fixed and variable-length conditions across ten different OpenML tasks. Notably, the `TOT` algorithm consistently outperformed all other methods, achieving the highest ROC AUC scores across all tasks. This suggests that `TOT`, which leverages the Tree of Thoughts approach, is particularly effective at navigating the search space and optimizing model parameters under variable conditions.

The comparison between variable-length and fixed-length methods revealed a mixed pattern. In Task 1, variable-length methods (`RS-V` and `BO-V`) achieved ROC AUC scores of 0.9129, outperforming fixed-length methods `RS-F` and `BO-F`, which scored 0.7706 and 0.7477, respectively. Similar trends were observed in Task 10, indicating the efficacy of variable-length methods in certain contexts. In Tasks 3, 7, and 8, fixed-length methods demonstrated superior performance. For instance, in Task 3, `RS-F` and `BO-F` achieved ROC AUC scores of 0.7153 and 0.7212, compared to 0.7039 and 0.7149 for `RS-V` and `BO-V`. This suggests that the benefits of variable-length optimization may be task-dependent.

Fixed-length methods like `RS-F` and `BO-F` optimize all parameters simultaneously. While this approach is straightforward, it may not efficiently navigate expansive and complex search spaces.

Table 1: ROC AUC OVO results of different optimization algorithms on the OpenML tasks. The best performance for each task is highlighted in bold.

| TASK | RS-F | BO-F | RS-V | BO-V | LLM-F | TOT |
|---|---|---|---|---|---|---|
| 1 | 0.7706 (0.0113) | 0.7477 (0.0117) | 0.9129 (0.0316) | 0.9129 (0.0316) | 0.742 (0.0006) | **0.9539 (0.021)** |
| 2 | 0.9977 (0.0016) | 0.9955 (0.0041) | 0.9954 (0.0015) | 0.9954 (0.0015) | 0.9737 (0.031) | **0.9978 (0.0016)** |
| 3 | 0.7153 (0.0073) | 0.7212 (0.0142) | 0.7039 (0.0169) | 0.7149 (0.0092) | 0.7213 (0.0115) | **0.7358 (0.0063)** |
| 4 | 0.768 (0.0156) | 0.7622 (0.0213) | 0.7572 (0.0327) | 0.7708 (0.0108) | 0.7419 (0.0063) | **0.7833 (0.0098)** |
| 5 | 0.7568 (0.0557) | 0.8113 (0.0248) | 0.7649 (0.0236) | 0.8057 (0.0092) | 0.8015 (0.0065) | **0.8205 (0.0187)** |
| 6 | 0.5512 (0.1226) | 0.7387 (0.0308) | 0.596 (0.1452) | 0.5745 (0.0948) | 0.727 (0.1029) | **0.7752 (0.0848)** |
| 7 | 0.885 (0.0503) | 0.9019 (0.0053) | 0.8755 (0.0409) | 0.8776 (0.038) | 0.8519 (0.0023) | **0.9125 (0.0098)** |
| 8 | 0.8573 (0.0211) | 0.8573 (0.0211) | 0.8397 (0.0009) | 0.8317 (0.0118) | 0.8607 (0.005) | **0.8594 (0.0091)** |
| 9 | 0.6354 (0.0166) | 0.6222 (0.0329) | 0.6251 (0.0303) | 0.6295 (0.0253) | 0.6258 (0.0168) | **0.6454 (0.0063)** |
| 10 | 0.8523 (0.119) | 0.9209 (0.0171) | 0.9212 (0.0034) | 0.9212 (0.0034) | 0.9274 (0.0047) | **0.9343 (0.004)** |

In contrast, variable-length methods require more resources to thoroughly explore the search space but can capture complexities that fixed-length methods might miss. The subpar performance of the variable-length LLM approach (`LLM-F`) emphasizes the challenges these methods face in high-dimensional spaces. In contrast, our `ToT` algorithm's sequential decision-making process allows it to effectively manage and exploit the complexities of variable-length optimization.

Furthermore, as shown in Figure 4, the evolutionary path taken by the `ToT` method during the optimization of the predictive model for Task 4 in OpenML illustrates its incremental improvement process. The step-by-step approach not only enhances overall predictive accuracy but also provides valuable insights into the contributions of each intervention along the way. By examining the improvements made at each stage, `ToT` proves to be a powerful tool for identifying the most impactful strategies for model enhancement. This insight is especially valuable in complex tasks, where understanding how various preprocessing and modeling techniques interact can be pivotal to achieving success.

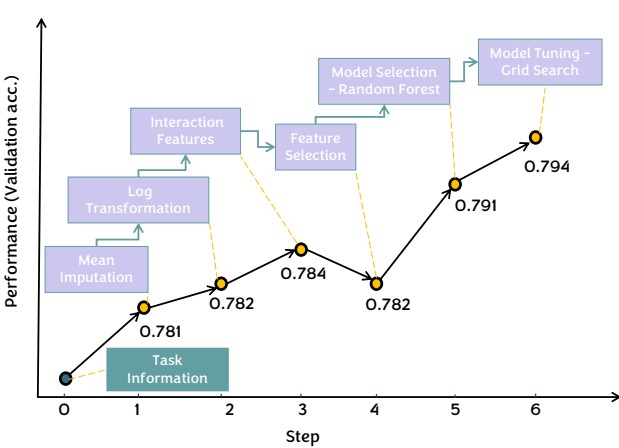

Figure 4: The evolutionary path of using `ToT` to solve the Task 4 in OpenML.

**Results on Clinical Datasets:** The results in Table 2 underscore the effectiveness of the `ToT` algorithm in clinical dataset optimization, consistently achieving top or near-top performance across multiple tasks. The varying success rates of other methods like `BO-V` and `LLM-F` suggest that while `ToT` generally provides superior performance, the optimal choice of algorithm may still depend on specific task characteristics or dataset nuances. Additionally, the variability in performance between fixed and variable-length methods across tasks indicates that variable-length optimization can be more effective but might require more nuanced implementation strategies to consistently outperform fixed-length approaches.

### 5.2 COMPARING WITH PEER COMPARISON

In this section, we present the results of our experimental evaluation, comparing the performance of our proposed `ToT` method against state-of-the-art AutoML frameworks such as `Auto-Sklearn`, `TPOT`, `H2O AutoML`, and `AutoKeras`, as well as traditional approaches like `RS`.

Table 2: ROC AUC OVO results of different optimization algorithms on the clinical tasks. The best performance for each task is highlighted in bold.

| TASK | RS-F | BO-F | RS-V | BO-V | LLM-F | TOT |
|------|------|------|------|------|-------|-----|
| 1 | 0.9311 (0.0096) | 0.9228 (0.0163) | 0.9235 (0.0167) | 0.9184 (0.0256) | 0.9333 (0.0096) | **0.9385 (0.003)** |
| 2 | 0.842 (0.0698) | 0.9275 (0.0195) | 0.831 (0.0268) | 0.8964 (0.0388) | NAN (NAN) | **0.9496 (0.003)** |
| 3 | 0.7438 (0.0147) | 0.6946 (0.0684) | 0.7566 (0.0098) | **0.7685 (0.0067)** | 0.7657 (0.0037) | 0.7605 (0.0018) |
| 4 | 0.8927 (0.0429) | 0.9209 (0.0166) | 0.9039 (0.0) | 0.9324 (0.0003) | 0.9328 (0.0006) | **0.9329 (0.0011)** |

Table 3: Performance metrics (ROC AUC) of peer algorithms on the OpenML tasks. The best performance for each task is highlighted in bold.

| TASK | RS | TOT | LLM-F | AUTO-SKLEARN | TPOT | H2O AUTOML |
|------|-----|------|-------|--------------|------|------------|
| 1 | 0.7706 (0.0113) | **0.9539 (0.021)** | 0.742 (0.0006) | 0.8222 (0.0275) | 0.8215 (0.0646) | 0.8771 (0.0152) |
| 2 | 0.9977 (0.0016) | 0.9978 (0.0016) | 0.9737 (0.031) | 0.9954 (0.0015) | **0.9982 (0.0018)** | 0.9942 (0.0015) |
| 3 | 0.7153 (0.0073) | **0.7358 (0.0063)** | 0.7213 (0.0115) | 0.7279 (0.0098) | 0.7209 (0.0101) | 0.7221 (0.0105) |
| 4 | 0.768 (0.0156) | 0.7833 (0.0098) | 0.7419 (0.0063) | 0.77 (0.0044) | 0.7639 (0.0194) | **0.7842 (0.0077)** |
| 5 | 0.7568 (0.0557) | **0.8205 (0.0187)** | 0.8015 (0.0065) | 0.8201 (0.0071) | 0.8142 (0.015) | 0.8087 (0.0093) |
| 6 | 0.5512 (0.1226) | **0.7752 (0.0848)** | 0.727 (0.1029) | 0.6293 (0.1272) | 0.5112 (0.0158) | 0.7586 (0.067) |
| 7 | 0.885 (0.0503) | **0.9125 (0.0098)** | 0.8519 (0.0023) | 0.9077 (0.0023) | 0.9094 (0.0052) | 0.8921 (0.0116) |
| 8 | 0.8573 (0.0211) | 0.8594 (0.0091) | 0.8607 (0.005) | **0.8682 (0.0157)** | 0.7609 (0.079) | 0.8488 (0.0114) |
| 9 | 0.6354 (0.0166) | **0.6454 (0.0063)** | 0.6258 (0.0168) | 0.6364 (0.0234) | 0.6272 (0.0318) | 0.6385 (0.0257) |
| 10 | 0.8523 (0.119) | **0.9343 (0.004)** | 0.9274 (0.0047) | 0.9141 (0.0096) | 0.9328 (0.0057) | 0.9326 (0.005) |

**Results on OpenML Datasets:** Table 3 shows the performance metrics for each method across the ten OpenML tasks. Our `ToT` approach outperformed all competitors in the majority of tasks, securing the highest ROC AUC scores in Tasks 1, 3, 5, 6, 7, 9, and 10. For example, in Task 1, `ToT` recorded an ROC AUC of 0.9539, surpassing Auto-Sklearn's 0.8222, TPOT's 0.8215, and H2O AutoML's 0.8771. This demonstrates the robustness of `ToT` in handling complex data distributions and optimizing model parameters effectively. However, `ToT` did not universally dominate. In Task 2, TPOT slightly outperformed `ToT` with an ROC AUC of 0.9982 versus 0.9978. Similarly, in Task 4, H2O AutoML edged out `ToT` with a score of 0.7842 compared to 0.7833, and in Task 8, Auto-Sklearn achieved the top score of 0.8682, demonstrating the competitive nature of these frameworks under certain conditions.

Table 4: Performance metrics (ROC AUC) of peer algorithms on the clinical tasks. The best performance for each task is highlighted in bold.

| TASK | RS | TOT | LLM-F | AUTO-SKLEARN | TPOT | H2O AUTOML |
|------|-----|------|-------|--------------|------|------------|
| 1 | 0.9311 (0.0096) | **0.9385 (0.003)** | 0.9333 (0.0096) | 0.5 (0.0) | 0.9352 (0.0082) | 0.9333 (0.0011) |
| 2 | 0.842 (0.0698) | **0.9496 (0.003)** | NAN (NAN) | 0.9447 (0.0018) | 0.9464 (0.003) | 0.9478 (0.0) |
| 3 | 0.7438 (0.0147) | 0.7605 (0.0018) | 0.7657 (0.0037) | 0.7522 (0.0099) | 0.6906 (0.0308) | **0.7665 (0.0045)** |
| 4 | 0.8927 (0.0429) | **0.9329 (0.0011)** | 0.9328 (0.0006) | 0.9319 (0.0003) | 0.9316 (0.0017) | 0.9326 (0.0014) |

**Results on Clinical Datasets:** Following the analysis of OpenML tasks, we extended our evaluation to clinical datasets to test each framework's capability in more sensitive and precision-critical applications. The performance metrics for these tasks are detailed in Table 4. `ToT` continued to show strong performance, achieving the highest ROC AUC in Tasks 1, 2, and 4. Notably, in Task 1, `ToT` achieved an ROC AUC of 0.9385, outperforming all other methods. Task 2 was particularly notable with `ToT` reaching a score of 0.9496, showcasing its efficacy in highly complex diagnostic tasks. However, in Task 3, H2O AutoML achieved the best result with an ROC AUC of 0.7665, slightly surpassing `ToT` which scored 0.7605. This indicates that while `ToT` is generally superior, other specialized frameworks can occasionally achieve better performance depending on the specific characteristics of the task.

**Results on Image Classification:** For the image classification task on Parkinson disease, we utilized the NAS-Bench-201 framework, applying our `ToT` method to optimize the connectivity among nodes. We compared our results with `RS`, `AutoKeras`, and the `Inception V3` architecture as reported in the original paper. The classification outcomes for this task are shown in Figure 5, which also serves to validate the efficiency of our proposed methods. The results highlight the effectiveness of the `ToT` approach in navigating the search space of network architectures more adeptly than traditional methods. By strategically determining connections between nodes, `ToT` not only enhances the performance of the model but also confirms its potential as a robust tool in AutoML for optimizing deep learning architectures. These findings underscore the capability of `ToT` to outperform established methods such as Inception V3 in specific tasks, suggesting that our approach can offer significant improvements in both efficiency and accuracy for complex image classification challenges.

Figure 5: Comparative results of the image classification task on Parkinson's disease using NAS-Bench-201 framework.

## 5.3 PARAMETER SENSITIVITY ANALYSIS

The maximum step length in the `ToT` method dictates the number of decisions made in constructing a model architecture before finalizing a configuration. Shorter step lengths tend to yield more conservative architectural choices, potentially leading to simpler models that may not fully capture the complexity required for certain tasks. On the other hand, longer step lengths facilitate a more thorough exploration of architectural possibilities, enhancing the model's sophistication but also heightening the risk of overfitting.

Our experiments, which varied the maximum step length, demonstrate the sensitivity of our system to this parameter. The results, detailed in Table 5, reveal that extending the maximum step length generally improves performance metrics across most tasks, with step lengths of 6 and 8 often yielding the best results. For instance, at a maximum step length of 6, tasks 1 through 5 show marked improvements in

Table 5: Performance metrics for varying maximum step lengths in the `ToT`.

| TASK | MAX. STEP=4 | MAX. STEP=6 | MAX. STEP=8 |
|------|-------------|-------------|-------------|
| 1 | 0.8751 (0.0682) | **0.9539 (0.021)** | 0.9254 (0.0679) |
| 2 | 0.9256 (0.0519) | **0.9978 (0.0016)** | 0.9921 (0.0062) |
| 3 | 0.7265 (0.0068) | **0.7358 (0.0063)** | 0.7320 (0.0115) |
| 4 | 0.7623 (0.0176) | **0.7833 (0.0098)** | 0.768 (0.0245) |
| 5 | 0.8167 (0.0218) | **0.8205 (0.0187)** | 0.8128 (0.024) |
| 6 | 0.6396 (0.0816) | 0.7752 (0.0848) | **0.7764 (0.0648)** |
| 7 | 0.891 (0.0021) | 0.9125 (0.0098) | **0.9144 (0.0067)** |

performance metrics compared to shorter step lengths. However, in task 6, it is the extension to a step length of 8 that provides a slight enhancement, indicating that the optimal step length may

vary by task. The trend suggests that while longer step lengths can lead to better exploitation of the model's capabilities, there is a nuanced balance between architectural complexity and overfitting, which varies depending on the task at hand.

## 6. CONCLUSION

In this paper, we formulated AutoML as a variable-length optimization problem, enabling dynamic adjustment of model architectures based on task complexity. Our approach, utilizing the tree of thoughts method in combination with LLMs, provides a flexible, efficient, and transparent exploration of the model space, adapting better to varying task demands. Experimental results across diverse datasets, including OpenML and clinical tasks, show that our method outperforms traditional fixed-structure AutoML systems. Particularly, the ToT approach demonstrates superior performance in both simple and complex tasks, consistently achieving higher ROC AUC scores across a wide range of tasks. Additionally, the transparency of the ToT method also allows for better interpretability, giving insights into how each decision impacts overall optimization. These advantages make the proposed method a powerful tool for model enhancement and adaptation, especially in domains where task complexity varies significantly. Future work will focus on expanding this framework to support more diverse machine learning tasks and applications, pushing the boundaries of AutoML in real-world settings.

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

## A. LITERATURE REVIEW

### A.1 AUTOML

Automated Machine Learning (AutoML) focuses on automating key stages of the machine learning pipeline, including data preparation, feature engineering, and hyperparameter optimization (Waring et al., 2020). Early AutoML frameworks such as Auto-WEKA (Thornton et al., 2013) and Auto-Sklearn (Feurer et al., 2019) focused on optimizing traditional machine learning pipelines, particularly for hyperparameter tuning. Recent tools like Auto-PyTorch (Zimmer et al., 2021) and AutoKeras (Jin et al., 2023) have expanded the scope to include neural architecture search (NAS) for deep learning models. Comprehensive AutoML solutions, such as Microsoft's NNI toolkit[1] and Vega, offer end-to-end pipelines, including data augmentation, NAS, model compression, and hyperparameter optimization. Google's AutoML suite[2] simplifies the process but still requires some level of user intervention. Despite these advancements, most current AutoML systems rely on fixed pipeline structures and predefined search spaces, limiting their flexibility to adapt to varying task complexities. Our method addresses this limitation by formulating AutoML as a variable-length optimization problem. This approach allows dynamic adjustment of model complexity, enabling flexible pipelines that better align with the complexity of individual tasks, in contrast to existing methods that operate within static frameworks.

### A.2 LLMS FOR AUTOML

LLMs have shown significant potential in enhancing machine learning tasks by autonomously decomposing and executing complex operations. They are increasingly recognized for their ability to provide convenient, comprehensive, and reliable decision-making across various applications. Several studies have directly explored GPT's capabilities in AutoML tasks such as feature engineering and NAS. For instance, GENIUS (Zheng et al., 2023) employs GPT-4 as a black-box optimizer to tackle NAS through an iterative refinement process. EvoPrompting (Chen et al., 2023) integrates LLMs as adaptive operators within an evolutionary NAS algorithm. Viswanathan et al. (Viswanathan et al., 2023) apply GPT-3.5 to achieve AutoML for specialized NLP models. In the realm of hyperparameter optimization (HPO), AutoMLGPT (Zhang et al., 2023) leverages LLMs to conduct HPO by iteratively prompting with data and model cards, mimicking model training via LLMs. However, this approach does not involve actual model training on real machines; instead, it relies on the zero-shot and few-shot learning capabilities of GPT models. Similarly, MLcopilot (Zhang et al., 2024) uses LLMs, informed by past experiences and knowledge, to predict optimal hyperparameter settings in a categorized manner. CAAFE (Hollmann et al., 2023) employs LLMs for automated feature engineering in tabular data, generating semantically meaningful features.

While these approaches demonstrate the potential of LLMs in automating machine learning tasks, they often rely on zero-shot and few-shot learning without incorporating iterative refinement based on real training performance. In contrast, our approach uses LLMs as dynamic agents capable of sequential decision-making within the AutoML framework. Our method not only automates complex ML operations but also optimizes them based on real training outcomes, enhancing the reliability and effectiveness of AutoML systems.

## B. DATASET

Table 6 and Table 7 summarize the tasks and their corresponding datasets.

---

[1] https://github.com/microsoft/nni
[2] https://cloud.google.com/automl

Table 6: Summary of OpenML Tasks and Corresponding Datasets.

| TASK | NAME | FEATURES | SAMPLES | CLASSES | OPENML ID |
|---|---|---|---|---|---|
| Task 1 | balance-scale | 4 | 125 | 3 | 11 |
| Task 2 | breast-w | 9 | 69 | 2 | 15 |
| Task 3 | cmc | 9 | 1473 | 3 | 23 |
| Task 4 | credit-g | 20 | 1000 | 2 | 31 |
| Task 5 | diabetes | 8 | 768 | 2 | 37 |
| Task 6 | tic-tac-toe | 9 | 95 | 2 | 50 |
| Task 7 | eucalyptus | 19 | 736 | 5 | 188 |
| Task 8 | pc1 | 21 | 1109 | 2 | 1068 |
| Task 9 | airlines | 7 | 2000 | 2 | 1169 |
| Task 10 | jungle-chess-2pcs | 6 | 2000 | 3 | 41027 |

Table 7: Summary of Clinical Tasks and Corresponding Datasets.

| TASK | NAME | FEATURES | TRAINING SET | TEST SET | CLASSES | DATA SOURCE |
|---|---|---|---|---|---|---|
| Task 1 | Metastatic disease [endocrinologic oncology] | 10 | 493 (34/66) | 295 (19/81) | 2 | Pamporaki et al. (2023) |
| Task 2 | Esophageal cancer [gastrointestinal oncology] | 105 | 7899 (3/97) | 6698 (2/98) | 2 | Gao et al. (2023) |
| Task 3 | Hereditary hearing loss [otolaryngology] | 144 | 1209 (76/24) | 569 (77/23) | 2 | Luo et al. (2021) |
| Task 4 | Cardiac Amyloidosis [cardiology] | 1874 | 1712 (50/50) | 430 (50/50) | 2 | Huda et al. (2021) |
| Task 5 | Parkinson disease | Image | 1097 (68/32) | 193 (68/32) | 2 | Wenzel et al. (2019) |

## C. Prompt

---

**Prompt for AutoML Operation Recommendation with ToT**

**Objective:** You are assisting with AutoML for a classification task using the {task} dataset from OpenML. Your goal is to generate an AutoML algorithm by recommending operations step-by-step based on the given search space, constraints, and characteristics of the dataset, using the **Tree of Thoughts (ToT)** approach. The recommendations should address data preprocessing, feature transformation, feature selection, model selection, and model tuning in a structured manner, following all constraints provided.

**Problem Analysis:**

- The shape of `X_train` is: {shape}. The categorical columns are: {categorical_columns}. If no categorical columns are present, skip encoding steps.
- The number of unique values in `y_train` is: {y_counts}.

**Search Space:**

- **Data Processing:** No Data Processing, Drop Missing Values, Replace Missing Values, Handle Outliers, Remove Duplicates, Mean Imputation, Median Imputation, Mode Imputation, KNN Imputation, Regression Imputation

- **Feature Transformation:** No Transformation, Min-Max Scaling, Z-score Standardization, L1 Normalization, L2 Normalization, Log Transformation, Square Root Transformation, Box-Cox Transformation, Polynomial Features, Interaction Features, PCA, LDA, One-Hot Encoding, Label Encoding

- **Feature Selection:** No Feature Selection, SelectKBest, SelectPercentile, SelectFromModel, RFE, Boruta, Feature Importance

- **Model Selection:** Logistic Regression, Decision Trees, Random Forest, XGBoost, SVM (Linear/RBF), MLP, CNN, Gradient Boosting Machines (GBM)

- **Model Tuning:** Grid Search, Random Search, Bayesian Optimization

**Constraints and Requirements:**

- Data Processing must always be the first step.
- Feature Transformation and Feature Selection must occur before Model Selection.
- Model Selection and Model Tuning must occur before training the final model.
- Each step must use an operation from the defined search space.
- You can recommend a maximum of {max_step} steps in total.

**Tree of Thoughts (ToT) Reasoning:**

- **Step 1:** Analyze the Dataset: Evaluate characteristics such as missing values, categorical variables, and task type (binary, multi-class). Consider any immediate data quality issues.

- **Step 2:** Propose Multiple Paths: Based on the current state and problem characteristics, propose multiple viable operations for the next step.

- **Step 3:** Explore Consequences: For each proposed path, consider the implications and how it impacts future steps. Which operation best addresses the current challenge, and how does it set up subsequent steps?

- **Step 4:** Choose the Best Path: Select the most promising operation after considering potential future steps. Ensure it adheres to constraints and is optimal for the current task.

**Example Completed Sequences:** ...
**Task:** Based on the {current_step_sequence}, recommend the next operation using ToT reasoning.
**Template for Response:**

- **Recommendation:** Choose the most appropriate method for the next step, considering dataset characteristics and the current step sequence.

- **Reasoning:** Propose multiple possible operations, explain the pros and cons of each, and justify the final choice based on how it aligns with future steps and constraints.

```
Next Step: [Select an appropriate method from the search space]
Reasoning: [Explain why this step was selected, considering dataset characteristics, future steps,
and the best path forward.]
```

---

Figure 6: The prompt used to generate operations for AutoML with ToT.

---

### Prompt for AutoML Pipeline Recommendation in Fixed-Length Optimization

**Objective:** You are assisting with AutoML for a classification task using the {task} dataset from OpenML. The goal is to generate an AutoML algorithm by recommending operations based on the given search space, constraints, and requirements. Recommendations should address data preprocessing, feature transformation, feature selection, model selection, and model tuning in a structured manner, adhering to the constraints provided.

**Problem Analysis:**

- The shape of `X_train` is: {shape}. The categorical columns are: {categorical_columns}. If there are no categorical columns, skip the encoding step.
- The number of unique values in `y_train` is: {y_counts}.

**Search Space:**

- **Data Processing:** No Data Processing, Drop Missing Values, Replace Missing Values, Handle Outliers, Remove Duplicates, Mean Imputation, Median Imputation, Mode Imputation, KNN Imputation, Regression Imputation
- **Feature Transformation:** No Transformation, Min-Max Scaling, Z-score Standardization, L1 Normalization, L2 Normalization, Log Transformation, Square Root Transformation, Box-Cox Transformation, Polynomial Features, Interaction Features, PCA, LDA, One-Hot Encoding, Label Encoding
- **Feature Selection:** No Feature Selection, SelectKBest, SelectPercentile, SelectFromModel, RFE, Boruta, Feature Importance
- **Model Selection:** Logistic Regression, Decision Trees, Random Forest, XGBoost, SVM (Linear/RBF), MLP, CNN, Gradient Boosting Machines (GBM)
- **Model Tuning:** Grid Search, Random Search, Bayesian Optimization

**Step-by-Step Reasoning (CoT):**

- **Step 1: Analyze the Dataset:** Assess the dataset to understand its structure and challenges. Consider the size of the dataset, the presence of missing values, outliers, or duplicates, and the types of features (numerical, categorical, or mixed). Also, evaluate the target variable to determine if it is a binary classification, multi-class classification, or regression task.
- **Step 2: Data Processing Options:** Propose appropriate data preprocessing steps based on the dataset characteristics.
- **Step 3: Feature Transformation and Scaling:** Recommend transformation or scaling techniques depending on the data distribution and the models that will be used.
- **Step 4: Feature Selection Options:** Select relevant features to optimize model performance.
- **Step 5: Model Selection:** Based on the problem type (classification or regression) and dataset characteristics, recommend appropriate models.
- **Step 6: Model Tuning:** Propose hyperparameter tuning methods and ensure that a cross-validation strategy, like k-fold cross-validation, is used to validate model robustness and avoid overfitting.
- **Step 7: Complete Pipeline Proposal:** After evaluating the dataset and possible operations, recommend the full AutoML pipeline. This should include data processing, feature transformation, feature selection, model selection, and tuning steps. Ensure that the proposed pipeline adheres to the constraints and is optimized for both the dataset characteristics and task requirements, ensuring the model is both efficient and accurate.

**Example of Completed Pipelines:**

- Input: `balance-scale`.
    - History 0: Mean Imputation, Log Transformation, Interaction Features, RFE, GBR; Model accuracy = 0.73.
    - ......

**Task:** Recommend the complete pipeline operations based on the given analysis.

**Template for Response:** 1. **Data Processing:** {your choice}  2. **Feature Transformation:** {your choice}  3. **Feature Selection:** {your choice}  4. **Model:** {your choice}  5. **Model Tuning:** {your choice}

Figure 7: The prompt used to generate AutoML pipeline in fixed-length optimization.

---

**Prompt for AutoML Code Generation**

You are an expert in AutoML and Python programming. I need your assistance in generating Python code to create an AutoML pipeline for a {task} task from the OpenML dataset. The pipeline should handle preprocessing, feature selection, model selection, and hyperparameter tuning.

**Objective**: For the {task} dataset from OpenML, generate Python code that builds an AutoML pipeline based on {current_step_sequence}. Ensure that the pipeline handles both classification and regression tasks dynamically, and that the pipeline can adjust based on the task's specific requirements (e.g., number of target classes).

**Requirements**:

- **Preprocessing**:
  - If `categorical_columns` is not empty, generate appropriate encoding for the categorical variables. Use `OneHotEncoder`, `LabelEncoder`, or another appropriate method.
  - If `categorical_columns` is empty, no encoding is necessary.
  - Handle missing values using strategies such as `SimpleImputer`.

- **Model Selection and Hyperparameter Tuning**:
  - If a model has already been selected in a previous step, use the current model for hyperparameter tuning and performance evaluation.
  - If no model has been selected, choose {n} models from the following list: Decision Trees, Random Forest, Gradient Boosting Machines (GBM), XGBoost, MLP.
  - Perform hyperparameter tuning for each of the selected models (or the current model) using Bayesian optimization, grid search, or random search. Selection of method can depend on available computational resources or task complexity.
  - Evaluate the performance of the model(s) based on 5-fold cross-validation using the appropriate metric:
    * For classification tasks, use ROC AUC (One-vs-One).
    * For regression tasks, use R2.
  - If multiple models were selected, choose the best model based on the evaluation results (e.g., highest cross-validation score).
  - Report the final performance of the selected model.

**Dataset Information**:

- `X_train, y_train = task.train_x, task.train_y`
- The shape of `X_train` is: {shape}
- The categorical columns are: {categorical_columns}
  - If `categorical_columns` is empty, skip encoding.
- The number of unique values in `y_train` is: {y_counts}

**Template for Response**:
```
Code: [Python Code]
Performance: [Final Cross-Validation Performance Value]
```

Figure 8: The prompt used to generate the code for AutoML based on current pipeline.

