# OpenReview forum: "Formulating AutoML as a Variable-Length Optimization Problem: A Tree of Thought Approach with LLM-Driven Code Generation"
_ICLR.cc/2025/Conference — ICLR 2025 Conference Withdrawn Submission_

### Official Review · Reviewer_6CjY · 2024-10-28

**Soundness:** 1
**Presentation:** 2
**Contribution:** 2
**Rating:** 3
**Confidence:** 5

**Summary:**

The paper proposes automated machine learning (AutoML) with variable-length optimization, an approach in which the exact number of machine learning (ML) pipeline components is not determined beforehand. That is, no static search space is given; instead, in this work, a maximal number of components shall be freely combined. To solve variable-length optimization, the paper proposes using a large language model (LLM) (with tree of thought) that determines which pipeline component to use next and produces the Python code to run the pipeline.

The proposed approach is motivated by the claim that differently complex tasks require a different length for the ML pipeline.
The paper claims that the proposed method enhances efficiency, improves transparency, and outperforms traditional AutoML systems.

**Strengths:**

## Flexible AutoML due to LLM-Driven Code Generation

The core strength of the paper is its proposed idea to solve AutoML without a fixed search space. As far as I can tell, the formulation based on variable-length optimization proposed in this paper is novel. Moreover, due to using an LLM with code generation, the potential search space and operators employed to solve the ML tasks are fully flexible (even if the prompt instructs the LLM to focus on a pre-defined set of operators).

## Agentic AutoML with Tree of Thought
Using a prompted LLM as an AutoML agent is a promising concept. Employing a tree of thought approach in this domain fits the problem well and is also original.

**Weaknesses:**

## Related Work & Baselines

This work makes several unsubstantiated and misguided claims about the AutoML field and AutoML systems. Moreover, it lacks a critical reflection of the current state of the art in AutoML. The short literature review in the appendix is severely limited and should be part of the main paper to put this work into greater context.

* To illustrate, the abstract claims that fixed pipeline structures "limit adaptability to diverse task complexities". A large fixed pipeline structure that relies on hyperparameter optimization to determine the best pipeline component (e.g., which model or which preprocessing) is, by definition, adaptable to diverse tasks. The failure of such an approach is more the cost of exploration during optimization and the inadequacy of HPO algorithms for large search spaces, as mentioned in Line 84. Likewise, the paper claims "[traditional AutoML] assumes that tasks share similar complexity levels" (Line 44), which, as far as I know, no AutoML systems ever assume as there would be no need for AutoML if there is free lunch. This motivation and research problem need to be more clearly defined, and the paper must clearly showcase how traditional AutoML fails to solve a task because it cannot adapt to the task at hand (instead of due to overfitting or other problems).
* The paper claims without references that AutoML systems like Auto-WEKA, H2O, and Auto-Sklearn 1.0 are mainstream (Line 39) and state-of-the-art (Figure 2, Line 298). Neither Auto-WEKA nor Auto-Sklearn are still mainstream nor state-of-the-art. H2O is a top contender but lacks modeling prowess compared to other systems. The paper especially ignores the current state-of-the-art, such as AutoGluon, MLJAR, LightAutoML, or FLAML. Please see the AutoML Benchmark [1] or the methods used by the top teams in Kaggle's AutoML Grand Prix [2] for current mainstream and state-of-the-art AutoML.
* The authors briefly reference TPOT  (Line 50) but do not differentiate their work from an evolutionary search process, which can also be seen as variable-length with pre-defined components. However, for the proposed method in this work, the components are also pre-defined by the prompt template (assuming the LLM abides by the prompt's instruction). Thus, I see a significant overlap between the formulation of variable-length optimization and any evolutionary search method (as common in NAS but less common in AutoML for tabular data). The authors also indicate this thought in Line 347 when speaking of an "evolutionary path". The distinguishing factor of this work is the code generation.

## Problem with the Method

While describing the method, the paper fails to mention several very important details. Moreover, some explanations in the method section prompt several alarming problems. In general, it is questionable how much of the method's performance comes from the idea or a multitude of confounding factors introduced by the implementation surrounding the core idea.

* The paper states that examples were added to the prompt (Line 210), which means the LLM performs in-context learning. This is very interesting, but from the prompt template in Figure 7, it looks like the examples come from the same task. So, how these examples were created and added to the prompt can significantly impact the performance, which requires a separate ablation study.
* In Line 234, the paper states, "If, after several iterations, the program still does not perform satisfactorily, we manually code the pipelines." - does this mean the authors manually verified that their AutoML systems worked on each test dataset and even adjusted the automatically generated code manually? Moreover, how did you do this for the test datasets? How often did this happen? This should never occur, nor should a manual intervention ever be allowed to evaluate how well the AutoML system works. This points to a major problem with the reproducibility and generalizability of the proposed method.
* Sections 3.3 and 3.4 are highly alarming and lack almost all the details required to explain them properly.  Are you doing HPO with a grid or a random search? What search space? What model library? These are all necessary details to explain the methods that are missing. Moreover, the procedure explained in Sections 3.3 and 3.4 is entirely missing in the overview Figure 3.
* In Section 3.3, the paper states that "for classification tasks or mean squared error for regression tasks" are used by the method. This should be the optimization metric (e.g., ROC AUC used later in the results, Line 308) and not hardcoded. The AutoML user must be able to specify the target metric. Likewise, the employed validation strategy (e.g., what kind of split) is very important to disclose here due to its impact on overfitting.

## Experimental Flaws

The tabular AutoML field luckily has a comprehensive and fair benchmark for comparing AutoML systems as introduced by the AutoML Benchmark [1]. This benchmark explains and abides by many important standards for comparing AutoML systems and tabular machine learning methods. Yet, this work seems to ignore almost all of this in favor of a questionable evaluation protocol. At the same time, the paper ignores the added complexity of rigorous scientific evaluation when using LLMs.

* The paper does not contain memorization tests for the tabular dataset used in the evaluation. As a result, it is impossible to determine from the paper if the results come from the LLM memorizing the best pipeline or querying it from OpenML or if it comes from its world knowledge. See [3] for an extended discussion. Furthermore, several of the datasets used in this paper have already been shown to be memorized by GPT-4 (the LLM used in this paper).
* Claims about the state-of-the-art cannot be made by evaluating only ten datasets for AutoML, where an extensive range of tasks must be solvable. I recommend using a more concrete selection procedure that clearly specifies the limits of the experiments and relying on curated benchmarking suites.
* The evaluation relies on repeated train-test splits, which is questionable for an evaluation strategy. Although this follows one prior work reference, for tabular machine learning and AutoML, 10-fold cross-validation is much more appropriate. This is done in the AutoML Benchmark or papers from other AutoML systems.
* One of the last experiments looks at the performance when varying the max step sizes, which shows drastic differences in performance based on step size. Based on this, how was the maximal step size for all experiments chosen? Did the authors take the best one based on Table 5 and used this for their results (it looks like this is the case based on the numbers in other tables)? In other words, did the authors tune this hyperparameter on the test data? This is problematic and requires a much more extensive study independent of the final test data.
* The paper fails to mention several details about how other AutoML systems were run, e.g., which metrics these methods optimize and how many resources they were given for the 1 hour. Depending on such settings, the AutoML systems might not be comparable. Likewise, it would be good to state how expensive it is to run this AutoML system with GPT-4.
* How were the descriptions mentioned in Line 273 created? Some descriptions on OpenML are manually created and not informative or correct. How do you manually create such a description without biasing the evaluation for a benchmark?
* The experiment in Section 5.1 is missing almost all the required information. What BO algorithms are used? What search space? What validation strategy? How do you train to perform variable-length surrogate models for BO? Neither the text nor the appendix allow the reader to understand what was compared here. Likewise,  in Line 377, what is the search space for RS? Why did you not use AutoKreas for tabular tasks?
* The NAS-Bench-201 experiments look very weird. The best pipelines for these are known on the internet and maybe to GPT-4, so how does the evaluation guard against this possibility? Furthermore, are the reported results correct? The scores for all metrics are identical, which seems very implausible.

## Failure to Provide Evidence for Claims

The paper claims that the proposed method enhances efficiency, improves transparency, and outperforms traditional AutoML systems.
Neither of these claims is supported by sufficient evidence in the paper.

* "outperforms traditional AutoML systems": Given the flaws in the experimental design, such a general statement is questionable at best.
* "enhances efficiency": The results never discuss improvements to efficiency (like time saved) but only focus on predictive performance.
* "Improves transparency": This claim confused me throughout the paper. The examples that explain better transparency (Figure 2 and Figure 4) are practically equivalent to an optimization trace, which also any other AutoML system can produce. That is, at what time did which model or hyperparameter improve the validation score is not a new form of transparency for an AutoML pipeline. Moreover, tree of thought does not cause this kind of transparency either.

## Minor
* Line 47, "trnasformation"
* In Equation 1, xi ∈ {Op1, Op2, Op3} needs to be a union of Op1, Op2, and Op3 instead.

# References
1. Gijsbers, Pieter, et al. "Amlb: an automl benchmark." Journal of Machine Learning Research 25.101 (2024): 1-65. (https://jmlr.org/papers/volume25/22-0493/22-0493.pdf#page=20.40)
2. AutoML Grand Prix 2024, https://www.kaggle.com/automl-grand-prix
3. Bordt, Sebastian, et al. "Elephants Never Forget: Memorization and Learning of Tabular Data in Large Language Models." arXiv preprint arXiv:2404.06209 (2024). (https://arxiv.org/abs/2404.06209)

**Questions:**

There are too many open questions to condense them. I recommend trying to reply to the points I have raised in the box above. To summarize, the main areas for questions and suggestions are:

* Put the paper into the greater context and current state-of-the-art of the AutoML field.
* Describe the proposed method's missing details, limitations, and confounding factors.
* Use an experimental setup that is sufficiently sophisticated for tabular machine learning and using LLMs trained on data from the internet (or that perform RAG with the internet).
* Support claims, made in the abstract and introduction, with results in the paper.

---

### Official Review · Reviewer_uGee · 2024-11-04

**Soundness:** 2
**Presentation:** 2
**Contribution:** 2
**Rating:** 3
**Confidence:** 4

**Summary:**

This paper presents an AutoML framework that reframes the AutoML pipeline as a variable-length optimization problem. Traditional AutoML methods use fixed-length pipelines, limiting their adaptability across tasks with differing complexities. The proposed method combines the Tree of Thoughts (ToT) approach with Large Language Models (LLMs) for dynamic pipeline generation. The paper conducted experiments on datasets from OpenML and clinical domains, which may require further analysis and explanation.

**Strengths:**

- The flexible AutoML framework uses variable-length optimization to adjust pipeline complexity based on the specific task requirements, increasing its adaptability and effectiveness.
- The ToT method enables the model to efficiently explore large search spaces by prioritizing optimal paths through a structured decision-making process, enhancing its navigation and effectiveness.
- This exploration could lead to new advancements in automated machine learning by leveraging the capabilities of LLMs.

**Weaknesses:**

- The experimental design and results lack clear, logical explanations, and no code has been provided to facilitate the reproducibility of these experiments. Additionally, several concerns have been raised regarding the validity and rationale of the experiments, as detailed in Questions 1-5.

- The study lacks comparisons with recent state-of-the-art methods, despite the availability of several comparative approaches and benchmarks within the field. It is recommended that the paper incorporates representative, cutting-edge methods from the past three years to more comprehensively validate the effectiveness of the proposed method.

- The structure of this paper is somewhat confusing, and the research process and key findings are sometimes difficult to understand, making it difficult to read. Additionally, certain technical details are insufficiently explained, potentially hindering readers’ understanding of the core content. Comparative methods are mentioned only by name, lacking essential descriptions.

- The manuscript contains several typos. For example, "feature trnasformation" in line 48 should be "feature transformation," and "dataset dataset" in line 195 should be corrected to "dataset." A thorough proofreading is needed.

**Questions:**

1. In Table 1, LLM-F shows the best performance rather than ToT on task 8, which means the bolding is incorrect. This suggests that ToT did not achieve SOTA performance across all tasks. Does the paper need to double-check all experimental results?

2. In Table 1, for tasks 1, 2, and 10, the experimental results for RS-V and BO-V are identical, including the values in parentheses. Moreover, the paper does not clarify whether these values represent variance or standard deviation. A similar occurrence is observed in task 8, where RS-F and BO-F also produce identical results. Are these results a mistake by the authors? If not, could the paper provide a more detailed analysis and explanation?

3. In Tables 2 and 4, the results indicate “NAN (NAN),” which is unclear. The paper previously mentioned that methods failing to produce results within an hour would not be compared. Could this mean that these results were not obtained within an hour? If so, I have three questions. First, since runtime varies across hardware and environments, what is the experimental platform? Are all experiments conducted on the same machine to ensure fairness? Second, why does the fixed-length LLM-F yield no results for task 2 while the variable-length TOT does? Finally, does “NAN (NAN)” imply that none of the five experimental trials produced results? Or does it mean that some of the five experiments did not produce results?

4. In Table 2, RS-V shows a standard deviation of 0 on task 4 across five trials (we assume that it is the standard deviation), implying highly consistent results. However, in other tasks, RS-V’s standard deviations are 0.0167, 0.0268, and 0.0098, so I don't think it can achieve a standard deviation of 10^{-5} for this task. Could you please provide more data to substantiate this?

5. In Table 4, Auto-Sklearn’s performance for task 1 is recorded as "0.5 (0.0)." Notably, while other methods score above 0.93 on this metric, Auto-Sklearn reaches only 0.5—an inconsistency not seen in other tasks. Furthermore, the standard deviation is 0.0. Both of these points remain unexplained in the paper.

6. The comparative methods used in this paper include Auto-WEKA (2013), H2O (2020), and Auto-Sklearn (2019), and more recently (e.g., 2022-2024) SOTA methods in the field are expected.

7. Lines 306-308 state, "... LLM for fixed-length approaches, and our proposed ToT for variable-length. Each method was tested under both fixed (RS-F, BO-F, LLM-F) and variable (RS-V, BO-V, ToT) conditions." This indicates that LLM-F is a fixed-length approach. However, lines 342-344 describe "The subpar performance of the variable-length LLM approach (LLM-F)... " suggesting LLM-F is instead a variable-length approach. Could the paper clarify this inconsistency?

8. This paper’s contribution centers on a variable-length AutoML process designed for broad applicability across diverse tasks. However, an essential question remains: how does the proposed method balance generality with accuracy? I am particularly interested in understanding whether the algorithm can outperform task-specific algorithms on complex or specialized tasks, such as those in regulated or medical domains. Could the paper consider providing additional experiments on the tasks with special scenarios to address this aspect?

---

### Official Review · Reviewer_fc9r · 2024-11-04

**Soundness:** 4
**Presentation:** 3
**Contribution:** 4
**Rating:** 8
**Confidence:** 4

**Summary:**

The paper "Formulating AutoML as a Variable-Length Optimization Problem: A Tree of Thought Approach with LLM-Driven Code Generation" presents a novel framework for Automated Machine Learning (AutoML) that adapts to task complexity by treating AutoML as a variable-length optimization problem. This approach differs from traditional AutoML methods, which rely on fixed pipelines, as it lets you adjust the model structure dynamically to better match various tasks.

The proposed framework leverages the “Tree of Thoughts” (ToT) approach along with Large Language Models (LLMs) to build machine learning models iteratively and sequentially. This means each decision in the model-building process is evaluated, allowing the models to gradually evolve with minimal manual effort. Moreover, LLMs also generate the necessary code for each step, turning model configurations into executable pipelines. In this way, it enhances transparency and reduces the need for human input.

Experimental results show that TOT outperforms conventional fixed-structure AutoML systems across various datasets (OpenML datasets and clinical ones) by achieving superior model performance and adaptability. Key contributions include formulating AutoML as a variable-length optimization problem and applying the ToT method with LLMs for efficient search space navigation.

**Strengths:**

* Originality: It's a novel contribution in the field of AutoML by rethinking the traditional fixed-pipeline structure. It's the first method to integrate the ToT method for designing AutoML pipelines. Using LLMs for incremental code generation and presenting the decision-making process in the model construction is useful in practive

* Quality: Good experimental design. The method is tested on multiple datasets (OpenML and clinical datasets) and against established AutoML algorithms (Auto-Sklearn, TPOT, H2O). It includes a maximum step length analysis.

* Clarity: Well-organized and logically structured paper. The figures and tables are effectively used to illustrate key concepts and experimental results.

* Significance: AutoML as variable-length optimization enables more adaptive model creation, and stimulate new research direction in the AutoML field with more focus on self-adjusting pipelines using language models.

**Weaknesses:**

Major points:
* Computation complexity due to LLMs. The computational footprint and latency can be substantial. Consider conducting a more detailed analysis of computational efficiency and resource requirements, and maybe include comparisons to existing AutoML methods.
* Lack of clarity on the decision-making process in ToT. It's not very clear how different thought paths are evaluated or pruned throughout the optimization. Consider adding a flow diagram or pseudocode showing how the ToT method selects and ranks decisions at each stage
* Lack of transparency on the code generation. "If after several iterations, the program still does not perform satisfactorily, we manually coding the pipelines” -> How often does this happen? As a suggestion: describe in more detail how code validation and error handling works and mention how often manually coding is needed.

Minor points:
* In the first part of the paper, you refer to OpenML as a single dataset, not a platform that contains a collection of datasets
* Figure 5 has overlapping ticks for the values of accuracy, precision, recall and f1-score

**Questions:**

See the Weaknesses section.

---

### Note · Authors · 2024-11-29

I have read and agree with the venue's withdrawal policy on behalf of myself and my co-authors.